



# Extreme wind turbine response extrapolation with Gaussian mixture model

Xiaodong Zhang[1] and Nikolay Dimitrov[1]

[1]Technical University of Denmark, Department of Wind and Energy Systems, Frederiksborgvej 399, 4000 Roskilde, Denmark

**Correspondence:** Nikolay Dimitrov (nkdi@dtu.dk)

**Abstract.** The wind turbine extreme response estimation based on statistical extrapolation necessitates using a minimal number of simulations to calculate a low exceedance probability. The target exceedance probability associated with a 50-year return period is $3.8 \times 10^{-7}$, which is challenging to evaluate with a small prediction error. The situation is further complicated by the fact that the distribution of wind turbine response might be multi-modal, and the extremes belong to a different statistical

population than the main body of the distribution. Traditional theoretical probability distributions, mostly uni-modal, may not be suitable for this task. The problem could be alleviated by applying a fit specifically on the tail of the distribution. Yet, a single uni-modal distribution may not be sufficient for modeling diverse wind turbine responses, and an inappropriate distribution model could lead to significant prediction errors, including bias and variance errors. The Gaussian mixture model, a probabilistic and flexible mixture distribution model used extensively for clustering and density estimation tasks, is infrequently

applied in the wind energy sector. This paper proposes using the Gaussian mixture model to extrapolate extreme wind turbine responses. The performance of two approaches is evaluated: 1) parametric fitting first and aggregation afterward, and 2) data aggregation first followed by fitting. Different distribution models are benchmarked against the Gaussian mixture model. The results show that the Gaussian mixture model is capable of estimating a low exceedance probability with minor bias error, even with limited simulation data, and demonstrates flexibility in modeling the distributions of varying response variables.

## 1 Introduction

An accurate low exceedance probability estimation is crucial in the statistical extrapolation of wind turbine responses, especially when limited data is available. The crude Monte Carlo Simulation (MCS) requires a large sample size, making it computationally expensive. The extreme response with a 50-year return period is usually extrapolated from 10-min simulations, and estimating the 50-year extreme response corresponds to an exceedance probability of $3.81 \times 10^{-7}$ from 10-min

simulations. Using crude MCS for analysis with such low probabilities requires at least $\frac{10}{3.81 \times 10^{-7}} = 26,280,000$ 10-min simulations for sufficient accuracy. Statistical extreme load extrapolation for wind turbines with normal turbulence load is required in IEC 61400-1 4th edition (IEC, 2019), where load extrapolation methods are categorized into two approaches, i.e., 1) parametric fitting first and aggregation afterward (ffaa); 2) data aggregation first and fitting afterward (affa). Unlike crude MCS, both approaches use limited data and rely on statistical extrapolation for low exceedance probability estimation, which will

introduce prediction error.





In the ffaa approach, the long-term distribution is obtained by aggregating the short-term distributions weighted by the probabilities of occurrence at different wind speed bins. The long term distribution $F_{\text{long term}}(s|T)$ of the load response $s$ with simulation time $T$ is a function of short term distribution $F_{\text{short term}}(s|V_k,T)$ at different wind speed (IEC, 2019):

$$F_{\text{long term}}(s|T) = \sum_{k=1}^{M} F_{\text{short term}}(s|V_k,T)p_k \qquad (1)$$

where $V_k$ is the mean wind speed at each wind bin, and

$$p_k = f(V_k)\Delta V_k, V_{\text{in}} \leq V_1 < ... < V_M \leq V_{\text{out}} \qquad (2)$$

is the probability of occurrence of each wind bin, with $f(V_k)$ being the probability density of the wind speed. $V_{\text{in}}$ is the cut-in wind speed (usually 3 m/s), $V_{\text{out}}$ is the cut-out wind speed (usually 25 m/s), and $\Delta V_k$ is the wind speed bin width (usually 2m/s). The wind speed is divided into bins within the wind turbine's operational range. At each wind speed bin, 10-min simulations are

performed, and the short-term distribution of extremes $F_{\text{short term}}(s|V_k,T)$ is fitted. The limitation of this approach is that even though $F_{\text{short term}}(s|V_k,T)$ might have different probabilistic behaviors at different wind speed bins, the same type of probability distribution is predefined and adopted for ease of application. The tail distribution might not be modeled well at all wind speed bins as there may be uncertainty in the individual fits of the underlying short-term distributions (Freudenreich and Argyriadis, 2008). The key issue is determining the proper distribution model without knowing the underlying conditional distributions at

each wind bin. Different distribution models have different tail behaviors, which could result in different long-term response predictions (Ding and Chen, 2013). The estimated extreme loads have a wide range when using the three-parameter Weibull distribution, Gumbel distribution, generalized extreme value (gev), and lognormal distribution as conditional distributions (Freudenreich and Argyriadis, 2008; Dimitrov, 2016). An improper distribution could result in a far-off extreme load prediction.

In the affa approach, the environmental input conditions for the 10-minute load simulations are sampled directly from

their long-term distributions. Hence the data aggregation happens automatically through the distribution choice. The extreme values from all 10-min simulations are fitted to a single probability distribution for extreme load estimation or exceedance probability calculation. This method is also referred to as the post-processing method (Zhang et al., 2020) and is equivalent to a density estimation approach. It suffers from the same challenge as the ffaa approach, as selecting a proper distribution becomes difficult, especially when the underlying distribution is unknown. An improper distribution selection could introduce

large estimation errors and far-off extreme response estimation. As the number of simulations is limited, even though different probability distributions are available, selecting the distribution is challenging. Fit accuracy at the center of the distribution does not guarantee a good extrapolation at the tail, which is more important for extreme load estimation. The wind turbine extreme response distributions could have multiple modals (Yang et al., 2022), whereas distributions like Weibull, Gumbel, gev, lognormal, etc., are uni-modal distributions. Fitting wind turbine extreme response with uni-modal distributions directly

will have a large estimation error at both the center and the tail distribution. Yang et al. (2022) compared the affa and ffaa approaches with different distributions but has not resolved the statistical extrapolation issue. A possible solution is to fit only the tail data, e.g., Natarajan and Holley (2008) fitted the tail of wind turbine loads using a Gumbel distribution with a quadratic distortion.





The Gaussian mixture model (GMM) (McLachlan and Peel, 2000) is proposed in this paper as the distribution function for
affa approach. It is a flexible probabilistic model and is widely used for machine learning tasks: clustering (He et al., 2011;
Zhang et al., 2021; Weber et al., 2022), classification (Huang et al., 2005; Kim and Kang, 2007; Permuter et al., 2006), and
image segmentation (Nguyen and Wu, 2013; Yin et al., 2018; Gupta and Sortrakul, 1998). GMM has also been used in the field
of wind energy, e.g., wind speed probability density estimation(Wahbah et al., 2018), wind power ramps(Cui et al., 2018), wind
turbine power (Zhang et al., 2019), wind turbine power curves (Srbinovski et al., 2021), and environmental contour estimation
(Zhang and Natarajan, 2022). GMM is a mixture model whose probability distribution function (pdf) is multimodal, which is
suitable for modeling the multi-modal distribution of wind turbine responses. However, its potential in wind turbine extreme
response estimation is yet to be explored. The objective of the present paper is to use GMM for extreme response estimation and
especially for response variables whose distribution is multi-modal. Comparison will be made against using other distributions
within the scope of the ffaa and affa approaches. The prediction error, which includes bias error (the difference between mean
prediction and true value) and variance error (the variability of the prediction), will be systematically investigated on different
wind turbine responses.

## 2   Gaussian mixture model

GMM is a weighted sum of Gaussian distribution components, where each component is defined by its mean ($\mu$) and standard
deviation ($\sigma$). The pdf of a GMM is

$$y(x) = \sum_{j=1}^{m} \pi_j \mathcal{N}(x|\mu_j, \sigma_j) \tag{3}$$

where $\mathcal{N}(x|\mu_j, \sigma_j)$ is the pdf of a Normal distribution, $m$ is the number of components, and $\pi_j$ is the component coefficient
(weight) and follows

$$\sum_{j=1}^{m} \pi_j = 1, \quad 0 \leq \pi_j \leq 1 \tag{4}$$

For a given number of components, the model parameters $\{\pi_j, \mu_j, \sum_j, j = 1, 2, ..., m\}$ could be estimated from a data sample
whose size is denoted by $N$, $\{x_n, n = 1, 2, ..., N\}$ (see Appendix A for more details). The number of components is unknown
a priori and should also be estimated from data. It balances the model to prevent underfitting or overfitting. The Akaike
information criterion (AIC) (Akaike, 1998) could be used for estimating the optimal number of components $m$. The method
stems from information theory and is an extension of a maximum likelihood estimation with the expression

$$\text{AIC} = 2k - 2\ell(\boldsymbol{\theta}|\boldsymbol{x}), \tag{5}$$

where $k = 3m - 1$ is the number of parameters. The $m$ and $\boldsymbol{\theta} = \{\pi_j, \mu_j, \sigma_j, j = 1, 2, ..., m\}$ that give the minimum AIC value
correspond to the optimal number of components and the associated model parameters respectively.

As a mixture model, GMM with two or more components is not Gaussian distributed anymore and has different tail behavior
compared to the Gaussian distribution. As its number of components and the associated component coefficients adapt to data,





GMM possesses more flexibility than other parametric distribution models. However, it's important to note that when the
sample size is small, too many components in GMM may result in an overrepresentation of the data and compromise its
extrapolation capability.

## 3    Problem formulation and methods

The wind turbine time domain analysis is time-consuming and requires a lot of computational resources, and the feasible
number of simulations is often limited. The low exceedance probability is usually extrapolated from a small sample. Statistical
extrapolation thus reduces the number of simulations but introduces prediction errors simultaneously. The prediction error
could be described by the $MSE$ of the estimator $\hat{P}_F$, i.e., $MSE(\hat{P}_F) = V(\hat{P}_F) + \left[B(\hat{P}_F)\right]^2$ (Wackerly et al., 2008), where
$V(\hat{P}_F)$ is the variance error, which describes the variability of prediction using different random samples, and $B(\hat{P}_F) =$
$E(\hat{P}_F) - P_F$ is the bias error, which describes the difference between the mean prediction with true value. The variance and
bias error, performance indicators of the wind turbine response extrapolation methods, should be examined.
Choosing an inappropriate distribution will undoubtedly increase the bias error in the ffaa and affa approaches. Furthermore,
even with the same distribution, different estimations of model parameters can yield varying results. Parameter estimation
includes but is not limited to 1) the method of least squares (ls), which estimates the model parameters by minimizing the sum
of the squares of the difference between the observed values and the predicted values from a model; 2) maximum likelihood
estimation (mle); and 3) method of moments (e.g., Weibull model parameter estimation (Moriarty et al., 2004)). In the ffaa
and affa approaches, a comparison is made between the gev, three parameter-Weibull, and lognormal distributions, along with
GMM using different parameter estimation methods. This comparison involves a total of 11 methods:

1. ffaa with gev, Weibull, and lognormal as short-term distributions(referred to as ffaa (gev), ffaa (Weibull), ffaa (lognormal)) conditional on wind speeds

2. affa with the three distributions using mle and ls for model parameter estimation (referred to as gev (mle), Weibull (mle),
lognormal (mle), gev (ls), Weibull (ls), lognormal (ls))

3. affa with GMM using AIC and ls for finding the number of components (referred to as GMM (AIC) and GMM (ls))

In the affa approach using mle, only the tail data (above $80\%$ quantile) is used to fit the theoretical distribution, from
which the probability of exceedance of the tail data is calculated as $Pt$. The probability of exceedance of the response variable is then determined as $P_f = \frac{N_t}{N} \times Pt = 0.2 \times Pt$, where $N_t$ is the sample size of the tail data, and $N$ is the total sample
size. The affa approach using ls minimizes the squared difference between the empirical and theoretical probability of exceedances in the logarithmic scale, with weights assigned to each data point. Specifically, for the sorted data sample in ascending order $S = \{x_i, i = 1, 2, ..., N\}$, the corresponding empirical cdf $\{Fe_i = (i - 0.5)/N, i = 1, 2, ..., N\}$, and the weight
$\{w_i = 1/\sqrt{Fe_i \times (1 - Fe_i)}\}$. The residual for the tail data $\sum_{i=N-N_t+1}^{N} w_i \times (\log{(1 - Fe_i)} - \log{(1 - F(x_i|\theta))})^2$ is minimized to get the model parameter $\theta$. Note that the empirical cdf is calculated based on the entire sample, but the squared error
is minimized only on the tail data.



Determining the appropriate number of components of a GMM requires further research. In Eq. 5, AIC is used (referred to as GMM (AIC)), where the first term is a penalty term, which discourages overfitting a model, thus balancing model complexity. However, in cases where the sample size is large, the penalty term becomes relatively small, and the AIC approaches maximum likelihood estimation. It will lead to better accuracy at the center of the distribution relative to the tail. To address this, ls on

tail data (above $80\%$ quantile) (legend as GMM(ls)) is proposed for selecting the number of components $m$. Following the procedure in section 2, the squares of the residuals from the tail data using GMM associated with the number of components $k$ are calculated (same as affa with ls). The optimal value of $m$ gives the least squares of residuals. Note that the ls here is only for determining the number of components, which is independent of the model parameter $\theta = \{\pi_j, \mu_j, \sigma_j, j = 1, 2, ..., m\}$ estimation using EM algorithm (see Appendix A).

The analysis focuses on four wind turbine responses: (a) the maximum out-of-plane blade tip deflection, (b) the maximum blade root out-of-plane bending moment, (c) the maximum blade root in-plane bending moment, and (d) the maximum tower base side-to-side bending moment. The simulated wind turbine responses are obtained from Barone et al. (2011). To extract peak responses from a sampled time series, the global maxima, block maxima (Fogle et al., 2008; Dai et al., 2022) and peak over threshold methods, average conditional exceedance rates (ACER)(Naess et al., 2013) could be used(Toft et al., 2011;

Dimitrov, 2016; Ding et al., 2013). Different peak extraction techniques will render different results but will not be the focus of this study, where the peaks are obtained using only the global maxima method. The randomness of the responses comes from the wind parameters, where the mean 10-minute wind speed is sampled from the Rayleigh distribution, and two random seeds are used for generating the turbulence model. The wind turbine model is a 5 MW NREL reference wind turbine, a three-bladed, upwind rotor with a diameter of 126 meters. The FAST aeroelastic code is used for the five million aero-elastic simulations

(Barone et al., 2011).

The 11 methods are firstly compared on the entire sample with a sample size $N \sim 10^5$. Those methods that exhibit relatively small differences compared to the results obtained through crude MCS are further assessed for prediction error, which involves performing statistical extrapolation using 100 sets of smaller sample size samples randomly drawn from the entire sample. Additionally, for the affa approach using the three distributions, the impact of varying amounts of tail data on the results will

also be discussed.

## 4 Results

The pdf of wind turbine response (d) from MCS is shown in Fig. 1, which is multimodal. To model the pdf of response (d) using the entire sample, the Weibull distribution with maximum likelihood estimation is employed. However, as the Weibull distribution is unimodal and cannot capture the multimodal nature of the underlying PDF, a noticeable discrepancy arises

between the MCS and the Weibull distribution. Consequently, the Weibull distribution cannot be directly used to model the wind turbine response (d). Similar observations can be made when utilizing other unimodal distributions such as gev, lognormal, and Gumbel distributions to model other responses. This limitation explains why, in the affa approach, only the tail portion of the distribution is fitted and utilized for extreme response estimation.





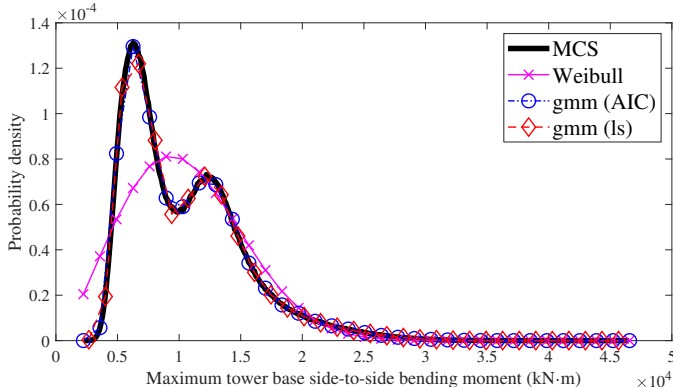

**Figure 1.** Pdf of maximum tower base side-to-side bending moment

## 4.1 Wind turbine response distribution modeling

The 11 methods are firstly compared on the entire sample with a sample size $N \sim 10^5$, and the probabilities of exceedance in logarithmic scale are plotted in Fig. 2. Since the predicted exceedance probability already reaches the limit that can be calculated using a Monte Carlo Simulation (MCS), this comparison focuses solely on statistical modeling rather than extrapolation.

In the ffaa approach, a large difference is observed between the prediction and MCS for cases (a) and (b) regardless of the short-term conditional distribution used. However, when using the gev distribution for cases (c) and (d) and the lognormal distribution for case(d), a smaller difference is observed. The results indicate that ffaa is a viable option for extreme response estimation, but distribution selection significantly impacts the results.

In the affa approach, using mle for model parameter estimation may not yield favorable outcomes. A significant difference between the prediction and MCS results is observed for all cases, regardless of the distribution used, except for case (b) when using the three-parameter Weibull distribution. Among the three distributions, Weibull consistently performs better than the others regarding tail performance. By focusing only on fitting the tail data and disregarding the accuracy of the distribution at the center, using the ls approach greatly improves the results. For all four cases, the difference between the prediction and MCS is small, except when using the lognormal distribution for case (d), as shown in Fig. 2 (d). While the choice of distribution has a relatively small effect on the results when using LS for the affa approach, it is important to note that an improper distribution selection can still lead to significant deviations in predictions.

Regarding GMM, using ls to determine the number of components outperforms AIC. When the sample size is large, the penalty term in AIC becomes negligible, often resulting in selecting a model with excessive components. In the four cases considered, ls suggests using $m = 6$ components, while AIC suggests using $m = 9$, explaining the difference in performance.

It is important to note that the results shown in Fig. 2 are based on the entire sample. In practical scenarios, only a relatively small number of simulations can be conducted, and the low probability of exceedance is statistically extrapolated. Since the results of the ffaa and affa approaches using mle for model parameter estimation heavily depend on the choice of distribution, the extrapolation methods employed will be compared in the subsequent section.





**Figure 2.** Probability exceedance estimation with MCS data

## 4.2 Wind turbine response distribution extrapolation

To statistically compare the extrapolation performance of different methods, a random sample of $10^4$ data points is drawn from
the entire sample 100 times. These samples are then utilized for extreme load extrapolation, and the statistical results obtained
from each method using these 100 samples are compared. With a sample size of $10^4$, the exceedance probability captured by
MCS will not be smaller than $10^{-4}$; thus statistical extrapolation performance of each method is compared. Given the relatively
large distribution modeling errors observed for the ffaa and affa with mle approaches, even when using the entire data sample,
their extrapolation performances are not compared here.

Fig. 3 shows results of affa approach with gev, three-parameter Weibll, and lognormal distributions with ls for model pa-
rameter estimation, along with GMM using ls and AIC for selecting the number of components. The mean probability of
exceedance from 100 samples of each method is plotted, and its difference with MCS represents the prediction bias error. The
gev, Weibull, and lognormal have relatively small bias errors for cases (a), (b), and (c), while relatively large bias errors for



case (d). Lognormal has the largest bias error for case (d), consistent with the observations made using the entire sample. On
the other hand, GMM has relatively small bias errors for all the cases, except case (d), when the exceedance probability is
smaller than $5 \times 10^{-6}$, where it exhibits a relatively larger bias error.

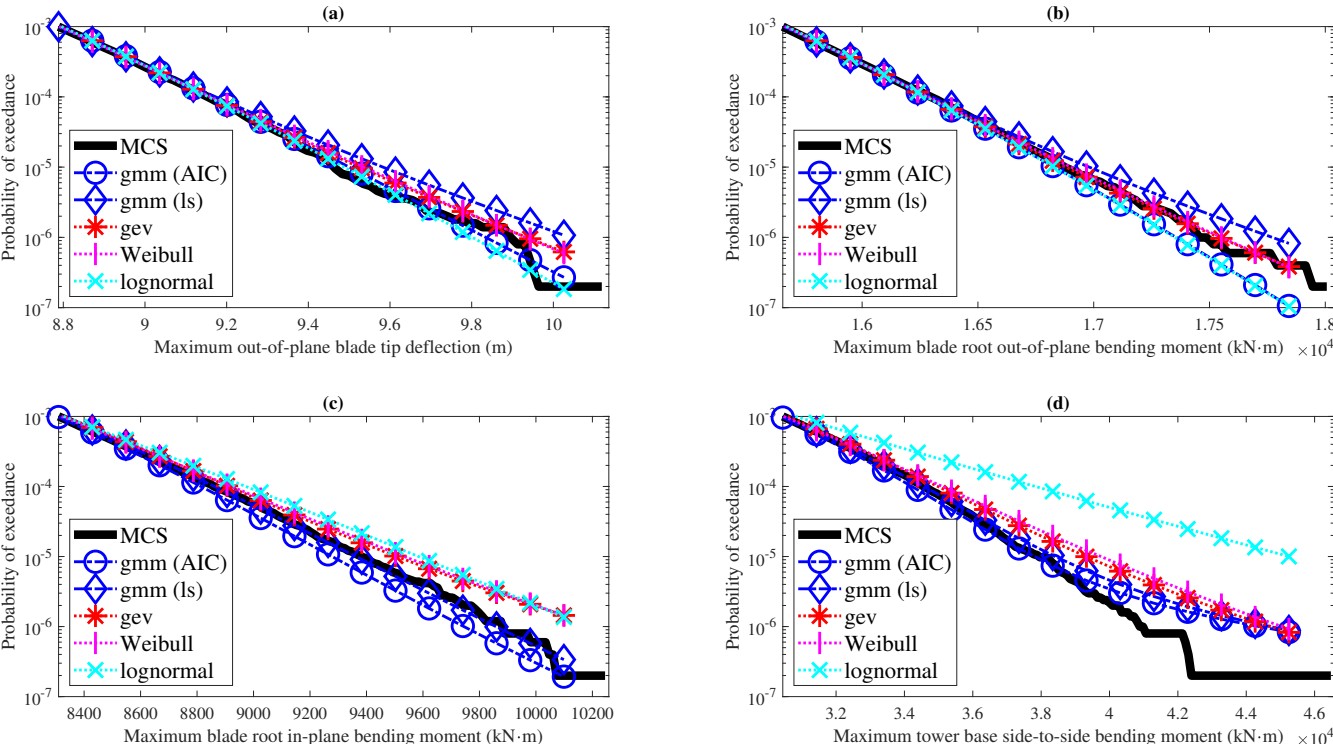

**Figure 3.** Probability of exceedance estimation with sample data

To make further comparisons, especially to examine the variance error of the methods compared, the prediction error at an
exceedance probability $P_F = 1.90 \times 10^{-6}$, the exceedance probability associated with a 10-year return period, is compared.
Assuming $\hat{P}_{F_{MCS}}$ is the true $P_F$, the associated extreme response values are obtained as (a) 9.75 m, (b) $1.74 \times 10^4$ kN·m, (c)
$9.77 \times 10^3$ kN·m, and (d) $4.03 \times 10^4$ kN·m for the four wind turbine responses. Several metrics are calculated and presented
in Table 1 to compare the performance of the methods at these response locations. These metrics include:

-   The mean of predicted exceedance probability, denoted as $E(\hat{P}_F)$

-   The variance of the predicted exceedance probability, denoted as $V$

-   The square of the bias error, denoted as $B^2$

-   The mean square error, denoted as $MSE$

-   the root mean square error (normalized by $P_F$), denoted as $rMSE/P_F$





These metrics provide a comprehensive evaluation of the methods in terms of their overall prediction error at the specific exceedance probability of interest.

**Table 1.** Extreme load estimation with sample data

|     |              | gev      | lognormal | Weibull  | GMM (AIC) | GMM (ls) |
|-----|--------------|----------|-----------|----------|-----------|----------|
| (a) | $E(\hat{P}_F)$ | 2.73E-06 | 1.47E-06  | 2.82E-06 | 1.78E-06  | 4.20E-06 |
|     | $V$          | 2.97E-11 | 3.04E-12  | 2.27E-11 | 1.09E-11  | 6.79E-11 |
|     | $B^2$        | 6.82E-13 | 1.87E-13  | 8.40E-13 | 1.43E-14  | 5.28E-12 |
|     | $MSE$        | 3.04E-11 | 3.23E-12  | 2.35E-11 | 1.10E-11  | 7.32E-11 |
|     | $rMSE/P_F$   | 2.90     | 0.94      | 2.55     | 1.74      | 4.50     |
| (b) | $E(\hat{P}_F)$ | 1.81E-06 | 9.46E-07  | 1.92E-06 | 9.70E-07  | 3.15E-06 |
|     | $V$          | 1.60E-11 | 1.26E-12  | 1.20E-11 | 1.82E-12  | 5.33E-11 |
|     | $B^2$        | 9.31E-15 | 9.16E-13  | 2.56E-16 | 8.70E-13  | 1.55E-12 |
|     | $MSE$        | 1.60E-11 | 2.17E-12  | 1.20E-11 | 2.69E-12  | 5.49E-11 |
|     | $rMSE/P_F$   | 2.10     | 0.77      | 1.82     | 0.86      | 3.89     |
| (c) | $E(\hat{P}_F)$ | 4.02E-06 | 4.83E-06  | 4.29E-06 | 8.72E-07  | 1.48E-06 |
|     | $V$          | 8.82E-11 | 1.88E-11  | 7.01E-11 | 7.49E-12  | 1.26E-11 |
|     | $B^2$        | 4.47E-12 | 8.57E-12  | 5.70E-12 | 1.06E-12  | 1.80E-13 |
|     | $MSE$        | 9.27E-11 | 2.74E-11  | 7.58E-11 | 8.55E-12  | 1.28E-11 |
|     | $rMSE/P_F$   | 5.06     | 2.75      | 4.58     | 1.54      | 1.88     |
| (d) | $E(\hat{P}_F)$ | 6.24E-06 | 4.59E-05  | 7.73E-06 | 3.04E-06  | 4.01E-06 |
|     | $V$          | 3.00E-10 | 2.72E-10  | 2.73E-10 | 1.51E-10  | 1.62E-10 |
|     | $B^2$        | 1.88E-11 | 1.94E-09  | 3.40E-11 | 1.30E-12  | 4.46E-12 |
|     | $MSE$        | 3.19E-10 | 2.21E-09  | 3.07E-10 | 1.52E-10  | 1.67E-10 |
|     | $rMSE/P_F$   | 9.38     | 24.71     | 9.21     | 6.48      | 6.78     |

Except for case (a), it can be observed from both Fig. 3 and Table 1 that the lognormal distribution exhibits a larger deviation in terms of the mean $\hat{P}_F$ compared to the gev and Weibull distributions. This discrepancy is indicated by the largest bias ($B^2$) value. However, when considering the variance error, the lognormal distribution shows the smallest mean square error ($MSE$) for the first three cases. This highlights the importance of considering variance errors when making comparisons. Conversely, the lognormal distribution yields the largest $MSE$ for case (d) among all the methods compared. This is due to the distinct tail behavior of case (d), which differs significantly from the lognormal distribution, as it is not sufficiently flexible to fit different response variables.

In contrast, the GMM demonstrates considerable flexibility. GMM (AIC) has the smallest $MSE$ for cases (c) and (d), and the second smallest for cases (a) and (b). On the other hand, employing the ls method to determine the number of components leads to more prediction errors than AIC in all four cases. This suggests that AIC performs well when the sample size is





relatively small. Table 2 provides the average number of components for GMM obtained from the 100 sets when using AIC
versus ls.

**Table 2.** Average number of components for GMM

|     | (a)  | (b)  | (c)  | (d)  |
| --- | ---- | ---- | ---- | ---- |
| AIC | 7.04 | 7.35 | 8.78 | 6.34 |
| ls  | 7.47 | 7.29 | 7.49 | 7.36 |

In almost all cases, the major contribution to the $MSE$ is $V$, as compared to $B^2$, which shows the importance of considering
variance error when performing a statistical comparison. As an exception, in case (d), the $B^2$ is larger than $V$, which indicates
that lognormal is unsuitable for this particular case as the bias error is excessively large.

### 4.3 Sample size effect on extreme load extrapolation

The sample size plays a significant role in determining the prediction error for all the methods examined. Van Eijkvan Eijk
et al. (2017) investigated the effect of different sample sizes on extreme load predictions and pointed out that the extrapolated
50-year responses could be misleading and a 300 min of time series is not sufficient for 50-year load extrapolation. In the
cases discussed above, a sample size of $10^4$ was used to achieve a relatively small prediction error close to the exceedance
probability associated with a 10-year return period. Limited by the sample size of the data, the exceedance probability smaller
than $1.90 \times 10^{-6}$ is not accurate for MCS and thus is not investigated in this study. Given the time-consuming nature of wind
turbine analysis, a smaller sample size is desirable. The results using 100 random sample sets with a size of $10^3$ are plotted in
Fig.4. In this analysis, GMM exhibits a relatively smaller bias for $P_F \geq 1.90 \times 10^{-5}$, associated with a 1-year return period.
On the other hand, for the other three distributions, the bias error becomes quite substantial for exceedance probability smaller
than $10^{-3}$. This demonstrates the limitation of all the compared methods when using a small sample size for extreme response
estimation.

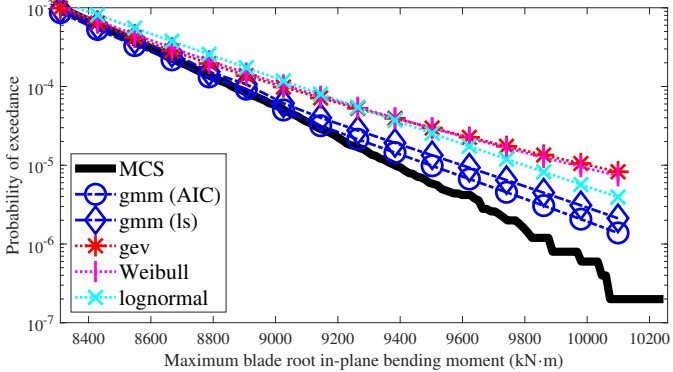

**Figure 4.** Results for case (c) with sample size $10^3$





## 5   Discussions

Statistical extreme response extrapolation is important for the probabilistic design of wind turbines. The ffaa and affa approaches in IEC 61400-1 (IEC, 2019) are assessed on their modeling and extrapolation performance, where the latter is a more
challenging task. The ffaa approach demonstrates feasibility when a suitable short-term distribution is selected, e.g., using gev as a short-term distribution for case (c) and (d) (as is shown in Fig. 2). However, it is important to note that the response at each wind speed bin may not follow the same distribution. Consequently, using a single distribution for all bins might introduce prediction errors. Furthermore, different response variables may exhibit different distribution characteristics, and applying a single distribution to various response variables may lead to additional prediction errors. As evident from Fig. 2, the gev might
perform well for cases (c) and (d) but not the other two response variables.

Flexibility in distribution modeling is important for both the ffaa and affa approaches. The results show that affa approach with ls is less subjected to the effect of choosing distribution models. However, an improper distribution will introduce prediction error. This could be seen from the examples above that using lognormal distribution for case (d) will have a large bias error. Given that wind turbine response variables can exhibit multi-modal distributions, it is recommended to utilize a flexible
distribution model. The Gaussian mixture model (GMM) fulfills this requirement, making it suitable for extreme response extrapolation. GMM offers the advantage of capturing the multi-modal nature of wind turbine responses and can provide more accurate predictions than other distribution models. Therefore, GMM is recommended for modeling and extrapolating extreme wind turbine responses.

There are several advantages to using the GMM for extreme response estimation:

1. GMM can effectively model both the center and the tail distribution. Figure 1 illustrates that GMM is capable of capturing the bimodal nature of the probability distribution well. In contrast, other uni-modal distributions struggle to model the pdf well, making them unsuitable for direct use in extreme response estimation.

2. GMM is a flexible modeling approach that can be applied to different types of wind turbine responses. It consistently performs well across all the compared wind turbine responses. In contrast, other distribution models may perform well
for certain response variables but exhibit significant prediction errors for others. For instance, as shown in Fig. 3 and Table 1, the lognormal distribution performs well for the first three cases but demonstrates the largest prediction error for the last case. Choosing an appropriate distribution model remains a challenge, and this issue also exists in the ffaa approach.

3. GMM eliminates the prediction error associated with selecting different percentages of data for tail estimation. Only the
tail data is utilized when fitting the three distributions using the ls method. However, the amount of tail data selected for estimation can lead to varying results. Figure 5 demonstrates the impact of using different amounts of data (data above different quantiles) on the results. GMM, fitted with ls, overcomes this limitation as ls is only used to determine the number of components and is not affected by the choice of quantiles.





In summary, the advantages of using GMM for extreme response estimation include its ability to accurately model both the
center and the tail distribution, its flexibility in handling different response variables, and its avoidance of prediction errors
associated with selecting different amounts of tail data. These benefits make GMM a recommended choice for wind turbine
extreme response extrapolation.

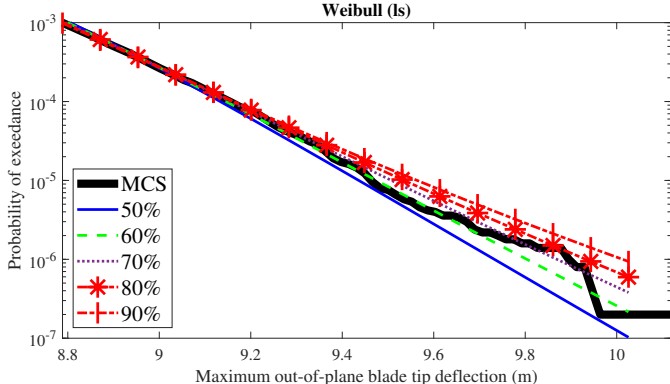

**Figure 5.** Results with different quantile

Statistical extrapolation is a challenging problem, particularly when dealing with smaller sample sizes for lower exceedance
probability extrapolation. With a sample size $10^4$, GMM has a noticeable bias error when the exceedance probability is smaller
than $5 \times 10^{-5}$ for case (d), as depicted in Fig. 3. It is important to note that when using a smaller sample size, the prediction
error is expected to increase, as illustrated in Fig. 4. Thus, caution must be exercised when performing statistical extrapolation,
and further research should focus on error analysis in statistical extrapolation. For instance, it would be valuable to investigate
the relationship between the sample size, the desired prediction error, and the extrapolated exceedance probability.
It should be mentioned that the favorable performance of GMM in the aforementioned examples is attributed to the un-
derlying distribution being multi-modal. For uni-modal variables, a flexible uni-variate distribution could be used for extreme
response estimation, e.g., the distribution based on maximum entropy with fractional moments (Zhang et al., 2020). Therefore,
the choice of an appropriate distribution should be guided by the characteristics of the variable being analyzed.

## 6 Conclusions

Extreme response estimation can be likened to low exceedance probability estimation with limited simulations. Both para-
metric fitting first and aggregation afterward and data aggregation first and fitting afterward approaches could be used for the
task. Both approaches with maximum likelihood estimation are quite sensitive to the distribution chosen, which could lead
to biased results with an improper probability distribution. The data aggregation first and fitting afterward coupled with least
square estimation for fitting the tail distribution is less sensitive to the type of probability distribution. However, an improper
probability distribution could still introduce a large prediction error.





There are flexible distributions available, but most are limited to uni-modal distribution. The probability distribution of the wind turbine responses could be multimodal. Using uni-modal distribution, e.g., gev, Weibull, and lognormal, to directly fit the distribution for extreme response estimation is infeasible. The Gaussian mixture model is a multimodal distribution by nature and is proposed here for extreme load estimation when the underlying distribution is multimodal. It could model both the

center and tail distribution, is flexible enough for different response variables, and does not require subjectively choosing the threshold compared with the least square estimation of tail distribution. It is thus recommended as an alternative distribution of wind turbine extreme response estimation.

Statistical extrapolation is challenging, and extreme caution must be exercised when making assumptions far beyond the available data sample. Given that the extreme response associated with a 50-year return period necessitates low exceedance

probability estimation, an adequate number of simulations are critical for accurate prediction.

*Code availability.* The code could be easily reproduced based on the description of the paper.

*Data availability.* The study used publicly available aero-elastic simulation data generated in previous work, for details please consult the authors in Barone et al. (2011) for details.

## Appendix A: GMM model parameter estimation

The initial model parameters are calculated from the clusters evaluated by the $k$-means clustering algorithm (Arthur and Vassilvitskii, 2007), and optimized by the Expectation-Maximization (EM) algorithm (McLachlan and Peel, 2000) as follows:

1. Divide the $N$ data points to $k$ clusters using $k$-means clustering algorithm. Compute $\mu_j$, $\sigma_j$ and $\pi_j$ using the data points within each cluster as the initial model parameters for the Expectation-Maximization (EM) algorithm

2. EM algorithm

The model parameters $\{\pi_j, \mu_j, \sigma_j, j = 1, 2, ..., m\}$ are found by an iterative EM algorithm (Dempster et al., 1977) to have a maximum likelihood estimation.

   (a) E step

   Evaluate the responsibilities using the current model parameters. The responsibility $\gamma_j(x_n)$ is the probability that component $j$ takes for explaining the observation $x_n$, which is calculated as:

$$\gamma_j(x_n) = \frac{\pi_j \mathcal{N}(x_n | \mu_j, \sigma_j)}{\sum_{i=1}^{m} \pi_i \mathcal{N}(x_n | \mu_i, \sigma_i)} \tag{A1}$$

   (b) M step



Update the model parameters using the responsibilities from E step. The mean for component $j$ is calculated as:

$$\mu_j = \frac{\sum_{n=1}^{N} \gamma_j(x_n) x_n}{\sum_{n=1}^{N} \gamma_j(x_n)} \tag{A2}$$

The standard deviation for component $j$ is calculated as:

$$\sigma_j = \sqrt{\frac{\sum_{n=1}^{N} \gamma_j(x_n)(x_n - \mu_j)^2}{\sum_{n=1}^{N} \gamma_j(x_n)}} \tag{A3}$$

and the $j$ component coefficient is calculated as:

$$\pi_j = \frac{1}{N} \sum_{n=1}^{N} \gamma_j(x_n) \tag{A4}$$

3. Repeat step 2 until the model parameters converge or the maximum number of iterations is met.

*Author contributions.* XZ developed the methodology with contributions from ND. XZ implemented the scientific methods. and validated the results. ND supervised the scientific work. XZ prepared the original draft. ND reviewed and edited the paper.

*Competing interests.* The co-author Nikolay Dimitrov is a member of the WES journal editorial board. The authors have therefore requested that the submission is handled by another editor.

*Acknowledgements.* This work has received funding from the Danish Energy Technology Development and Demonstration Program, EUDP, under the project ProbWind, with grant agreement 64019-0587. The authors appreciated that Professor Lance Manuel (The University of Texas At Austin) generously shared the aero-elastic simulation dataset link with Xiaodong Zhang in 2017 when the data was publicly available online.



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
