# Peer review of "Extreme wind turbine response extrapolation with Gaussian mixture model"

_Wind Energy Science, 2023_

## Referee Comment (RC1)

The paper approaches the topic of wind turbine extreme load statistical extrapolation by proposing the use of a multi-modal distribution. The characteristics of the distribution are compared against othe common choiches with various fitting methods. The topic is discussed clearly and the paper is well organized. I have some minor comments.

Introduction: Authors did a good job (in my opinion) to provide reference to previous work. I think the importance of the topic could be stressed better in more general standpoint for a less experienced reader. Providing a bit more context regarding how IEC61400-1 prescribes to employ statistical extrapolation techniques in the context of DLC calculations is advised.

I find the acronyms a bit confusing. I would advise to use only capital letters for them. Also strongly advise to include a nomenclature section.

Section 3: I cannot understand in which fitting methods and for which distributions only the tail data (above 80% quantile) is used to fit the distributions. I have understood that in the case of gmm the entire population is used. Please rephrase this section (or parts of it) to improve clarity in this regard.

L257: "Choosing an appropriate distribution model remains a challenge, and this issue also exists in the ffaa approach." As in compared to affa? Please explain more clearly.

Figure 5: there seems to be quite some difference between the curves here. The main body of text does not reflect this in my opinion. Could you elaborate?

---

## Referee Comment (RC2)

**Review**

**General comment:**
The main objective of the paper is to present a comparison of the wind turbine response in a Gaussian mixture distribution function and common unimodal distribution functions. The paper is well organized, and the distribution procedure is described clearly.

There are general notes as follows:

- The nominated distributions for the comparison include just three general distributions. The process or reason behind these distributions is not clear. As GMM has several parameters, it is expected that it has better GoF (Goodness of Fit) compared to simple two- or three-parameter distributions. In order to have a fair judgment, it is expected that mixture probability density functions such as GEV-Weibull or Weibull-Weibull are used in the comparison process. A list of such distributions is presented in the paper by Jung ( http://dx.doi.org/10.1016/j.enconman.2016.12.006 ).

- The authors used two random seeds in simulations. Different random seed numbers have an effect on the response of the turbine and, consequently, on the distribution of the extreme loads. That is why the IEC has a recommendation for the minimum number of seeds at different wind speeds. The number of seeds should be justified at least by a reference or with sensitivity analysis, as it is in contrast with IEC regulation for ultimate analysis (Annex G of IEC).

- There are several places where authors claim a statement without a related reference. Some of the examples are stated in specific comments.

- As the authors used load data from a previous publication, it is worth mentioning the DLCs that are included in the referenced publication in order to clarify the load's condition for the reader.

- As a reader, in the result section, the superiority of using GMM is not established clearly. For example, it seems the results in Table 1 are, to some extent, close.

**Specific comment:**
1. In Section 1, line 20, the sentence "*Using crude MCS for analysis with such low probabilities requires at least …*" needs to be referenced.
2. In Section 1, line 43, the sentence "*An improper distribution could result in a far-off extreme load prediction*" needs to be referenced. Is there any study that shows how much the results change with improper distribution?
3. In Section 1, line 54, the sentence: "*Fitting wind turbine extreme response with uni-modal distributions directly 55 will have a large estimation error at both the center and the tail distribution*" needs to be referenced.
4. In Section 3, line 96, it doesn't mention what MSE stands for. It is mentioned later in Section 4.2, but for readers, it should be cleared the first time it is used.

---

## Author Comment (AC1)

**Authors' response to comments of reviewers**

| | |
|---|---|
| Journal: | Wind Energy Science |
| Title of paper: | Extreme wind turbine response extrapolation with Gaussian mixture model |
| Authors: | Xiaodong Zhang, Nikolay Dimitrov |
| Manuscript No.: | wes-2023-69 |

**Authors' Response to the Comments of Reviewer #1**

The authors would like to thank the reviewer for the comments and advice on the submission. The manuscript has been revised accordingly, and the detailed responses are provided below.

5   **Overview**: *The paper approaches the topic of wind turbine extreme load statistical extrapolation by proposing the use of a multi-modal distribution. The characteristics of the distribution are compared against other common choices with various fitting methods. The topic is discussed clearly and the paper is well organized. I have some minor comments.*

**Response**: Your review of the manuscript and providing valuable comments are appreciated. The issues highlighted are addressed, and changes have been made to the revised manuscript. (The line numbers in the changes sections are the number in 10   the revised manuscript.)

**Comment 1**: *Introduction: Authors did a good job (in my opinion) to provide reference to previous work. I think the importance of the topic could be stressed better in more general standpoint for a less experienced reader. Providing a bit more context regarding how IEC61400-1 prescribes to employ statistical extrapolation techniques in the context of DLC calculations is advised.*

15   **Response 1**: Thanks for your comments.

**Changes 1**: In line 22, "The ultimate design load assessment procedure prescribed by the IEC aims at ensuring the structural integrity of the turbine when subjected to rare extreme loading conditions. The standards assume three types of scenarios for simulating such rare events: 1) extreme environmental conditions that result in extreme loads, 2) occurrence of faults potentially combined with extreme environmental conditions, and 3) rare occurrences under normal operation. The last option is repre-20   sented by Design Load Case (DLC) 1.1. It encompasses loads resulting from site-specific atmospheric turbulence occurring during the turbine's normal lifetime, i.e., the normal turbulence model. It establishes the characteristic load value corresponding to a 50-year return period, which could be obtained by statistical analysis of extreme loading using load extrapolation methods. The design load is then obtained by multiplying the characteristic loads by an appropriate partial safety factor. (IEC, 2019)"

**Comment 2**: *I find the acronyms a bit confusing. I would advise to use only capital letters for them. Also strongly advise to include a nomenclature section.*

**Response 2**: Thanks for your comments.

**Changes 2**: A "Nomenclature" section is added at line 306, and the capital letters are used for acronyms in the text and figure labels.

**Comment 3**: *Section 3: I cannot understand in which fitting methods and for which distributions only the tail data (above 80 % quantile) is used to fit the distributions. I have understood that in the case of gmm the entire population is used. Please rephrase this section (or parts of it) to improve clarity in this regard.*

**Response 3**: Thanks for the comments, in the LS estimated AFFA approaches, GEV (LS), Weibull (LS), and Lognormal (LS), only the tail data (above $80\%$ quantile) is used to fit the theoretical distributions.

**Changes 3**: In line 117, "This comparison involves a total of 11 methods:

1. FFAA with GEV, Weibull, and lognormal as short-term distributions(referred to as FFAA (GEV), FFAA (Weibull), FFAA (Lognormal)) conditional on wind speeds, where the MLE is used for model parameter estimation.

2. AFFA with the three distributions using MLE and LS for model parameter estimation (referred to as GEV (MLE), Weibull (MLE), Lognormal (MLE), GEV (LS), Weibull (LS), and Lognormal (LS))

3. AFFA with GMM using AIC and LS for finding the number of components (referred to as GMM (AIC) and GMM (LS))

In the LS estimated AFFA approaches, GEV (LS), Weibull (LS), and Lognormal (LS), only the tail data (above $80\%$ quantile) is used to fit the theoretical distributions, where the probability of exceedance of the tail data is calculated as $Pt$."

**Comment 4**: *L257: "Choosing an appropriate distribution model remains a challenge, and this issue also exists in the ffaa approach." As in compared to affa? Please explain more clearly.*

**Response 4**: The authors would like to state that the theoretical distributions are not flexible, and choosing a specific distribution for a response variable whose underlying distribution is unknown is challenging. For FFAA, as from section "4.1 Wind turbine response distribution modeling", choosing either of the three distributions will give a large prediction error. For AFFA, certain distributions may perform well for certain response variables but exhibit significant prediction errors for others. For instance, as shown in section "4.2 Wind turbine response distribution extrapolation", the lognormal distribution performs well for the first three cases but demonstrates the largest prediction error for the last case.

**Changes 4**: As the paragraph is intended to explain the flexibility of GMM, the sentence has been deleted to avoid confusion.

**Comment 5**: *Figure 5: there seems to be quite some difference between the curves here. The main body of text does not reflect this in my opinion. Could you elaborate?*

**Response 5**: Thanks for your comments.

**Changes 5**: In line 271, "The extreme load estimation for the maximum out-of-plane blade tip deflection is compared using
55 Weibull(LS) with data above the 50th to 90th quantiles. Fig. 5 illustrates significant differences when different amounts of tail
data are used."

The authors thank the reviewer for the comments and advice on the submission. The manuscript will be revised accordingly and the detailed responses are provided below.

60 **General comment:**

**Overview**: *The main objective of the paper is to present a comparison of the wind turbine response in a Gaussian mixture distribution function and common unimodal distribution functions. The paper is well organized, and the distribution procedure is described clearly.*

**Response**: Your review of the manuscript and providing valuable comments are appreciated. The issues highlighted are ad-
65 dressed and changes have been made in the revised manuscript. (The line numbers in the changes sections are the number in the revised manuscript.)

**Comment G.1**: *The nominated distributions for the comparison include just three general distributions. The process or reason behind these distributions is not clear. As GMM has several parameters, it is expected that it has better GoF (Goodness of Fit) compared to simple two- or three-parameter distributions. In order to have a fair judgment, it is expected that mixture proba-*
70 *bility density functions such as GEVWeibull or Weibull-Weibull are used in the comparison process. A list of such distributions is presented in the paper by Jung (http://dx.doi.org/10.1016/j.enconman.2016.12.006).*

**Response G.1**: Thanks to the reviewer for pointing out the additional methodologies with mixture probability distributions, they add to the discussion and we have included a short comment to inform the readers about this alternative. Given that the paper already has a large number of methodologies tested, we think that including additional options may make the paper
75 less readable and hence have decided not to make calculations with mixture distributions. In our view, applying mixture distributions could have benefits for modeling the entire population, but when only interested in the tail, the performance will likely be similar to the tail-only fits using the least-squares approach with a single distribution, because the tail is often dominated by the behavior of one of the underlying distributions.

Fig. 1 illustrates the PDF of response (c) in place of response (d) to showcase the possibility of the wind turbine response being
80 multimodal.

**Changes G.1**: In line 65, "Mixture probability distributions (e.g. Weibull-Weibull), as discussed in Jung and Schindler (2017), can be beneficial for accurately modeling an entire multimodal statistical population. In the tail of the response, the model will likely be dominated by only one of these distributions; hence the tail behavior predicted by a mixture distribution model is expected to be similar to that of a unimodal distribution fit focused on tail data only."

85 In line 158: The PDF of wind turbine response (c) from MCS is shown in Fig. 1, which is multimodal.

In line 166: the Fig. has changed to wind turbine response (c);

**Comment G.2**: *The authors used two random seeds in simulations. Different random seed numbers have an effect on the response of the turbine and, consequently, on the distribution of the extreme loads. That is why the IEC has a recommendation for the minimum number of seeds at different wind speeds. The number of seeds should be justified at least by a reference or with sensitivity analysis, as it is in contrast with IEC regulation for ultimate analysis (Annex G of IEC).*

**Response G.2**: We would like to clarify that we did not perform the simulations, and the authors from Barone et al. (2011) generated the simulation data, which was 96 years of simulations and computationally demanding in 2012. We agree that more seeds should be used, but it is computationally challenging for 96 years of simulations even today.

**Comment G.3**: *There are several places where authors claim a statement without a related reference. Some of the examples are stated in specific comments. As the authors used load data from a previous publication, it is worth mentioning the DLCs that are included in the referenced publication in order to clarify the load's condition for the reader.*

**Response G.3**: DLC 1.1 is used in the referenced publication.

**Changes G.3**: In line 150, "The FAST aeroelastic code is used for the five million aero-elastic simulations (Barone et al., 2011), which is based on design load case (DLC) 1.1 in IEC 61400-1 (IEC, 2019)."

**Comment G.4**: *As a reader, in the result section, the superiority of using GMM is not established clearly. For example, it seems the results in Table 1 are, to some extent, close.*

**Response G.4**: Yes, In terms of prediction error for a specific application, the superiority of GMM over other distributions is marginal. Nevertheless, the key advantage of employing GMM lies in its consistent performance owing to its high flexibility. GMM tends to exhibit favorable performance for diverse applications. While the lognormal distribution excels in three out of four compared scenarios, a notable prediction error is anticipated if the lognormal distribution is chosen for case (d).

**Changes G.4**: In line 223, "The distinguishing factor between GMM and other distributions lies in its performance consistency. Given the uncertainty about the underlying data distribution, opting for a flexible distribution with reliable performance, like GMM, becomes advantageous as it mitigates the risk of significant prediction errors caused by an inappropriate model selection."

**Specific comment:**

**Comment S.1**: *In Section 1, line 20, the sentence "Using crude MCS for analysis with such low probabilities requires at least ..." needs to be referenced.*

**Response S.1**: Thanks for the comments.

**Changes S.1**: In line 18: "The extreme response with a 50-year return period is usually extrapolated from 10-min simulations, and estimating the 50-year extreme response corresponds to an exceedance probability $p = 3.81 \times 10^{-7}$ from 10-min simulations. The coefficient of variation (c.o.v.) of the MCS estimator is $\sqrt{(1-p)/(pN)}$, where $N$ is the sample size(Ditlevsen

and Bjerager, 1986). Using crude MCS for analysis with such low probabilities requires at least $\frac{10}{3.81 \times 10^{-7}} = 26,280,000$ simulations for sufficient accuracy, i.e., a c.o.v. $\approx \sqrt{1/10} \approx 0.316$."

**Comment S.2**: *In Section 1, line 43, the sentence "An improper distribution could result in a far-off extreme load prediction" needs to be referenced. Is there any study that shows how much the results change with improper distribution?*

**Response S.2**: In Freudenreich and Argyriadis (2008), "Because of the load-influencing effect of the wind turbine control system, the fit of the distribution functions to the data points was difficult. The obtained loads differed up to $25\%$, depending on the fitting quality." Both Freudenreich and Argyriadis (2008) and Dimitrov (2016) showed the difference in extreme load prediction due to the different distributions used.

**Changes S.2**: In line 51, the reference is added: An improper distribution could result in a far-off extreme load prediction (Freudenreich and Argyriadis, 2008; Dimitrov, 2016).

**Comment S.3**: *In Section 1, line 54, the sentence: "Fitting wind turbine extreme response with unimodal distributions directly will have a large estimation error at both the center and the tail distribution" needs to be referenced.*

**Response S.3**: The sentence is not referenced from previous work but is based on this study, so it is moved to the Results section.

**Changes S.3**: The sentence is moved to line 163.

**Comment S.4**: *In Section 3, line 96, it doesn't mention what MSE stands for. It is mentioned later in Section 4.2, but for readers, it should be cleared the first time it is used.*

**Response S.4**: Thanks for your comments. 'MSE' stands for mean squared error.

**Changes S.4**: "mean squared error" has been added before the first 'MSE' in line 107.

**References**

Barone, M. F., Paquette, J. A., Resor, B. R., and Manuel, L.: Decades of wind turbine load simulation, No. SAND2011-3780C, 2011.

Dimitrov, N.: Comparative analysis of methods for modelling the short-term probability distribution of extreme wind turbine loads: Methods for modelling the probability distribution of extreme loads, Wind Energy, 19, 717–737, https://doi.org/10.1002/we.1861, 2016.

140 Ditlevsen, O. and Bjerager, P.: Methods of structural systems reliability, Structural Safety, 3, 195–229, https://doi.org/10.1016/0167-4730(86)90004-4, 1986.

Freudenreich, K. and Argyriadis, K.: Wind turbine load level based on extrapolation and simplified methods, Wind Energy, 11, 589–600, https://doi.org/10.1002/we.279, 2008.

IEC: International standard IEC61400-1: Wind turbines - part 1: design guidelines, Fourth edition, 2019.

145 Jung, C. and Schindler, D.: Global comparison of the goodness-of-fit of wind speed distributions, Energy Conversion and Management, 133, 216–234, https://doi.org/10.1016/j.enconman.2016.12.006, 2017.

---

## Author Response (AR2)

**Authors' response to comments of associate editor**

| | |
|---|---|
| Journal: | Wind Energy Science |
| Title of paper: | Extreme wind turbine response extrapolation with Gaussian mixture model |
| Authors: | Xiaodong Zhang, Nikolay Dimitrov |
| Manuscript No.: | wes-2023-69 |

The authors would like to thank the associate editor for the comments and advice on the submission. The manuscript has been revised accordingly, and the detailed responses are provided below.

**Overview**: *This paper discusses the topic of wind turbine extreme load statistical extrapolation methods. By utilizing the Gaussian mixture model (GMM) as a multi-modal statistical model to fit the long-term distribution of load response under uncertain wind condition, the data aggregation first and fitting afterward (AFFA) method can be used to estimate the characteristic load with 50-year return period. The performance of the proposed method is validated against the existing methods through four cases: the maximum out-of-plane blade tip deflection, the maximum blade root out-of-plane bending moment, the maximum blade root in-plane bending moment, and the maximum tower base side-to-side bending moment. A comparison of the fitting first and aggregation afterward (FFAA) and AFFA approach is conducted, and the advantages of GMM in terms of both accuracy and flexibility is demonstrated. The paper is concise and well organized with most of the reviewers' comments properly addressed. I would recommend this paper to be accepted on condition that the following minor issues are further addressed to improve the clarity and completeness of the paper:*

**Response**: Your review of the manuscript and providing valuable comments are appreciated. The issues highlighted are addressed, and changes have been made to the revised manuscript. (The line numbers in the changes sections are the number in the revised manuscript.)

**Comment 1**: *The discussion regarding Figure 5 is missing in the main body of the context.*

**Response 1**: Thanks for your comments.

**Changes 1**: The discussion regarding Figure 5 is updated at line 272:... can lead to varying results. GMM (LS) overcomes this limitation as LS is only used to determine the number of components and is not affected by the choice of quantiles. For example, in Fig. 5, the extreme load estimation for the maximum out-of-plane blade tip deflection is compared using Weibull(LS) with data above the 50th (legend as 50%) to 90th (legend as 90%) quantiles. Significant differences are observed when using different amounts of tail data for LS with Weibull distribution.

Fig. 5 is positioned closer to the text describing it.

**Comment 2**: *Regarding the Comment G.1 of Reviewer #2, in Line 68-71 of the revised manuscript, the authors claim that the tail distribution of a mixture probability distribution is likely to be dominated by only one of the distributions. This implies that the tail distribution of GMM might also be dominated by a single Gaussian distribution component since GMM is also a kind of mixture probability distribution. I am not sure this explanation is the true reason that the GMM outperforms other mixture distribution. Consequently, I recommend the authors discuss more about the advantage of GMM over other mixture distribution models or compare the performance of other mixture distribution models in Sections 3-5.*

**Response 2**: Thanks for your comments. The advantages of GMM over other mixture distribution models (e.g., the bi-modal distributions compared in Jung and Schindler (2017)) are listed as follows:

– Wind turbine extreme load responses can exhibit multiple modes, as illustrated in Figure 1 of the revised manuscript. When employing maximum likelihood estimation for the entire dataset, Weibull distribution struggles to accommodate distributions with more than one mode, as evidenced in Figure 1. Consequently, distributions like GEV-Weibull or Weibull-Weibull might not yield optimal performance in cases with multiple modes.

– The primary emphasis of the paper lies in load extrapolation, a scenario often extending beyond the available dataset, rather than on distribution modeling. The suitability of bi-modal distributions compared in Jung and Schindler (2017) for load extrapolation or estimating low failure probabilities has not been established.

– In Jung and Schindler (2017), various bi-modal distributions were presented, introducing the challenge of selecting an appropriate distribution when the true distribution of the underlying data remains unknown. As demonstrated in this manuscript, one of the advantages of GMM is its flexibility, mitigating prediction errors stemming from improper distribution selection.

**Changes 2**: In line 67: ...modelling an entire multimodal statistical population. However, the varying number of modes in wind turbine response distributions, sometimes exceeding two, poses challenges for both bi-modal distributions and fixed-component distributions. These challenges stem from mismatches in mode numbers during distribution fitting using the complete dataset. When relying solely on tail data for distribution fitting, the selection of appropriate mixture probability distribution components and their quantities remains a challenge (e.g., Weibull-Weibull, GEV-Weibull, GEV-Lognormal-Weibull).

**Comment 3**: *The word "sample" is used throughout the paper with various meanings, which is a bit confusing.*

**Response 3**: Thanks for your comments.

**Changes 3**:

In line 92, 130: data sample -> dataset

In line 107, 133, 154, 156, 161: sample -> dataset

**References**

55  Jung, C. and Schindler, D.: Global comparison of the goodness-of-fit of wind speed distributions, Energy Conversion and Management, 133, 216–234, https://doi.org/10.1016/j.enconman.2016.12.006, 2017.